# COVID-19 and systemic lupus erythematosus genetics: A balance between autoimmune disease risk and protection against infection

Yuxuan Wang[1], Suri Guga[1], Kejia Wu[1], Zoe Khaw[1], Konstantinos Tzoumkas[1], Phil Tombleson[2], Mary E. Comeau[3], Carl D. Langefeld[3], Deborah S. Cunninghame Graham[1], David L. Morris[1]ᐱ*, Timothy J. Vyse[1]ᐱ

1 Department of Medical & Molecular Genetics, King's College London, London, United Kingdom, 2 NIHR GSTFT/KCL Biomedical Research Centre, London, United Kingdom, 3 Department of Biostatistics and Data Science and Center for Precision Medicine, Wake Forest University School of Medicine, Winston-Salem, North Carolina, United States of America

ᐱ These authors contributed equally to this work.
* david.l.morris@kcl.ac.uk

**Data Availability Statement:** The data supporting this article is openly available from the King's

## Abstract

Genome wide association studies show there is a genetic component to severe COVID-19. We find evidence that the genome-wide genetic association signal with severe COVID-19 is correlated with that of systemic lupus erythematosus (SLE), having formally tested this using genetic correlation analysis by LD score regression. To identify the shared associated loci and gain insight into the shared genetic effects, using summary level data we performed meta-analyses, a local genetic correlation analysis and fine-mapping using stepwise regression and functional annotation. This identified multiple loci shared between the two traits, some of which exert opposing effects. The locus with most evidence of shared association is *TYK2*, a gene critical to the type I interferon pathway, where the local genetic correlation is negative. Another shared locus is *CLEC1A*, where the direction of effects is aligned, that encodes a lectin involved in cell signaling, and the anti-fungal immune response. Our analyses suggest that several loci with reciprocal effects between the two traits have a role in the defense response pathway, adding to the evidence that SLE risk alleles are protective against infection.

## Author summary

We observed a correlation between the genetic associations with severe COVID-19 and those with systemic lupus erythematosus (SLE, Lupus), and aimed to discover which genetic loci were shared by these diseases and what biological processes were involved. This resulted in the discovery of several genetic loci, some of which had alleles that were risk for both diseases and some of which were risk for severe COVID-19 yet protective for SLE. The locus with most evidence of shared association (*TYK2*) is involved in interferon production, a process that is important in response to viral infection and known to be dysregulated in SLE patients. Other shared associated loci contained genes also involved in

College London research data repository, KORDS, at https://doi.org/10.18742/19758484.

**Funding:** YW (reference No. 202008060031), SG (reference No. 201908330377), and KW (reference No. 201806100004) were funded by the King's-China Scholarship Council. The King's-China Scholarship Council funded YW, SG, and KW's PhD. The funders had no role in study design, data collection and analysis, decision to publish, or preparation of the manuscript. None of the authors received a salary from any funder for this study.

**Competing interests:** The authors have declared that no competing interests exist.

the defense response and the immune system signaling. These results add to the growing evidence that there are alleles in the human genome that provide protection against viral infection yet are risk for autoimmune disease.

## Introduction

The outbreak of COVID-19 together with modern genotyping technologies has given us the unprecedented opportunity to investigate the genetics of response to viral infection. Recent GWAS of severe COVID-19 have shown that there is a genetic component to the variability of the clinical outcome [1]. Some of the genetic loci identified unsurprisingly point to pathways involved in the host immune response. Therefore, a comparison between the genetics of severe COVID-19 and autoimmune disease (AID) may be enlightening. In this study we compare the genetics of severe COVID-19 with those of systemic lupus erythematosus (SLE). The rationale for selecting SLE is twofold: some SLE risk alleles act to augment the interferon response (e.g. *IRF5*, *IRF7*, *CXORF21-TASL*); other lupus susceptibility genes act in the intracellular viral sensing (e.g. *IFIH1*, *TLR7*, *RNASEH2C*) pathway.

## Results

### Genetic correlation

To investigate the shared genetics between SLE and severe COVID-19, we ran a genome-wide genetic correlation analysis between ancestry matched SLE and severe COVID-19 association data. The SLE data comprised a meta-analysis of three European GWASs [2–4] ($N_{cases}$ = 5,734, $N_{controls}$ = 11,609, **Table A in S1 Text**) and for the COVID-19 we used the GenOMICC release 1 European data [1] (critically ill patients with COVID-19 vs. ancestry-matched control individuals from UK Biobank, $N_{cases}$ = 1,676, $N_{controls}$ = 8,380, **Table A in S1 Text**). We found the two traits to be genetically correlated ($r_g$ = 0.56, s.e. = 0.16, p = 3 x $10^{-04}$). To identify which regions were driving this correlation we ran a local genetic correlation analysis that included Immunochip European data [5] in the SLE meta-analysis (additional $N_{cases}$ = 3,568, $N_{controls}$ = 11,245, **Table A in S1 Text**). This identified multiple loci with both positive and negative correlation of which the *TYK2* locus was the most significantly correlated (p-value = 1 x $10^{-04}$, **Table B in S1 Text**). This gene encodes a kinase that regulates transduction of IFN-I signaling. An overview of GWAS data used in the study is illustrated in **Fig 1** and Table A in S1 Text.

### Shared genetic associations: Severe COVID-19—SLE meta-analyses

To search for shared associations between SLE and severe COVID-19, we used summary association data from the large SLE meta-analysis (three SLE European GWASs plus immunochip), and for severe COVID-19 we used summary association data from the COVID-19 Host Genetics initiative (COVID-19 hg) [6] release 6 data (GenOMICC study is a subset of these data) association results of very severe respiratory confirmed COVID-19 vs. population (A2_ALL_leave_23andme, $N_{cases}$ = 8,779, $N_{controls}$ = 1,001,875, **Table A in S1 Text**).

We checked published associations in each trait to our summary association data for the other trait, and validated by coclisation analysis (coloc [7]), to identify potential shared risk loci (see Material and Methods). This found evidence of shared association at *TYK2*. Our colocalization analysis of all loci that had at least one SNP with p < 1 x $10^{-05}$ in both diseases (see Material and Methods) identified *TYK2* and *CLEC1A*, a C-type lectin that is a negative regulator of dendritic cells.

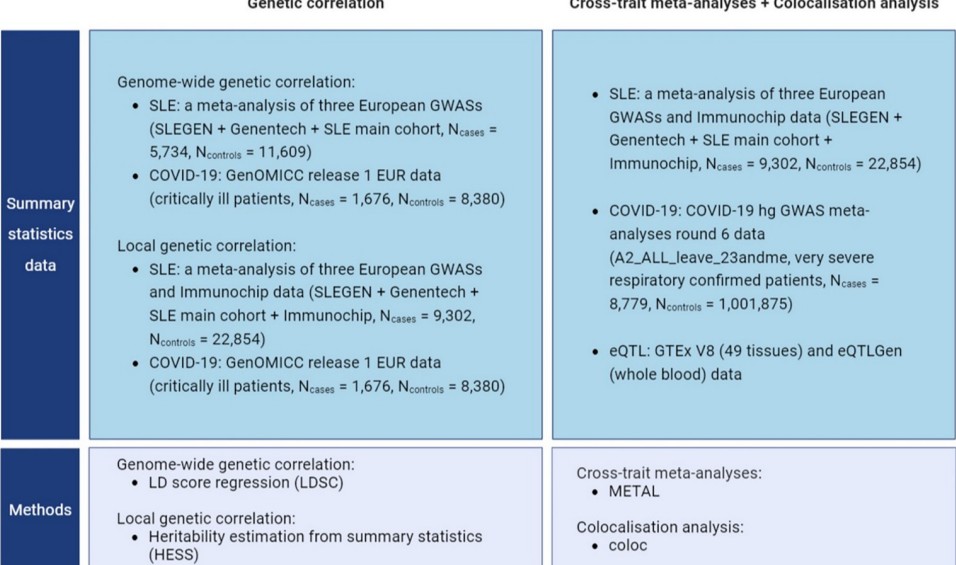

**Fig 1. Overview of GWAS data used in the study.**

We performed a cross-trait meta-analysis that included an analysis to highlight opposing effects (see Material and Methods, overlapped $N_{SNPs}$ = 1,559,546). Manhattan plots from the meta-analyses can be seen in **Fig 2**. There were 15 loci that had genome-wide significant evidence of (p-values < 5 x $10^{-08}$, **Table 1**), the very significant p-values at the *TYK2* locus in the lower plot (**Fig 2**) highlights the negative correlation at this locus. There were six association signals in five of these loci with colocalization probabilities ($PP_{H4}$) greater than 0.8 and three of these, implicating *CLEC1A*, *TYK2* and *PDE4A*, had $PP_{H4}$ > 0.95 (**Table 1**). The *TYK2-PDE4A* locus had opposing direction of effect across the two diseases and the other 4 loci (*CLEC1A*, *IL12B*, *PLCL1-RFTN2*, and *MIR146A*) had agreement in direction of effect. Though genome-wide significant evidence were found in the other 10 loci, there was relatively weak evidence for colocalization. Two well-known SLE associated loci, *IRF8* and *TNFSF4*, showed evidence of significant association in the opposing effect meta-analysis with some evidence for colocalisation of shared signals at both loci (*IRF8* $PP_{H4}$ = 0.36, *TNFSF4* $PP_{H4}$ = 0.37; **Tables C and D and Fig A in S1 Text**). LocusZoom plots for all other loci can be seen in **Figs B-L in S1 Text**. A pathway analysis showed that there was an enrichment of genes in defense response, cytokine-mediated signaling and type I interferon signaling pathway with over half the genes being included in one or more pathways (**Table E and Fig M in S1 Text**).

## Tyrosine kinase 2 (TYK2)

The *TYK2* locus has previously been found to be associated with SLE [4,8–12] and severe COVID-19 [1]. There was significant negative local genetic correlation (p-value = 1 x $10^{-04}$, ρ-HESS, overlapped $N_{SNP}$ = 2,544) at *TYK2* between the two diseases. In a stepwise regression approach using summary meta-analysis data for both traits, we found a highly significant overlap between genetic association signals (overlapped $N_{SNP}$ = 4,720); importantly, the SLE risk alleles were protective against severe COVID-19. The locus-wide association signals in COVID-19 and SLE are compared in **Fig 3A**. There were two independent signals that colocalized across traits (posterior probabilities of coloc = 0.991 and 0.993), referred to arbitrarily as signal-A and signal-B in **Table 2**. The top two SNPs independently associated with SLE

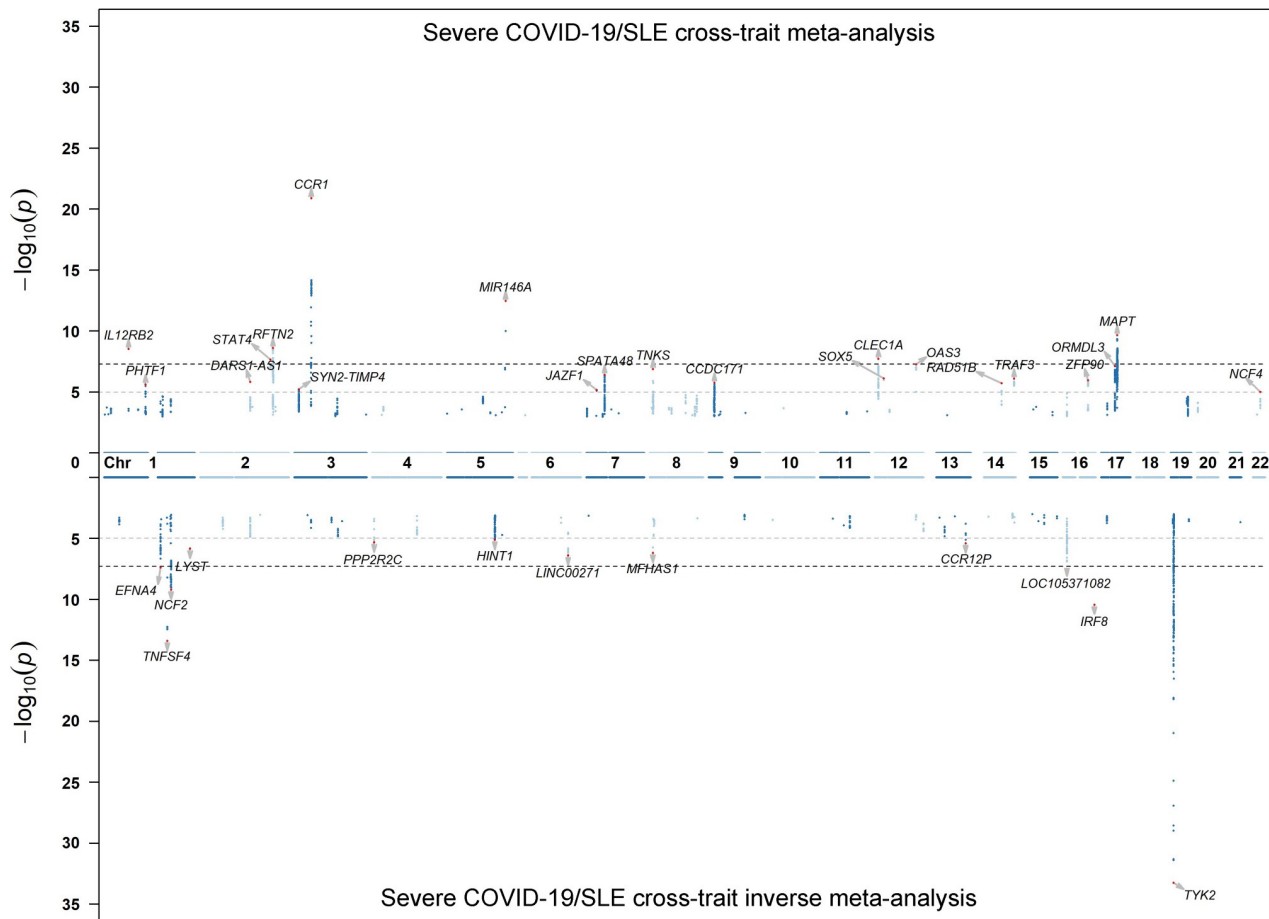

**Fig 2. Cross-trait meta-analysis log10 p-values.** Upper plot has results from a standard inverse variance meta-analysis. The lower plot has results form a meta-analysis when reversing the severe COVID-19 direction of effect. The MHC extended region (chr6: 24–36 Mb) was removed. Signals that the lead SNP has p-values $< 1 \times 10^{-05}$ were annotated by the closest gene names according to base pair position.

(rs34536443 and rs34725611) are in high LD ($r^2 = 0.88$ and 0.97 respectively), with the two SNPs we found to be independently associated with severe COVID-19 (rs74956615 and rs11085727) being reported previously in a COVID-19 GWAS [1]. For a full set of association results for these SNPs across traits see **Table F in S1 Text,** where it is shown that for all SNPs the effects have reciprocal directions of effect in SLE and COVID-19 outcome. In both traits, the relatively rare variants rs34536443/rs74956615 were associated independently from the more common variants rs34725611/rs11085727 (see conditional results bJ and pJ in **Table 2A and 2B**; $r^2 = 0.06$ and 0.09 between rs34536443 and rs34725611 and between rs74956615 and rs11085727 respectively in the EUR SLE data, $r^2 = 0.08$ and 0.07 in the 1000 genomes EUR data).

The lead SLE SNP rs34536443 for signal-A is a missense variant (**Table 2C**) and homozygosity at the SLE protective allele (C) drives a near complete loss of *TYK2* function and consequently impairs type I IFN, IL-12 and IL-23 signaling [13]. The genetic association in signal-A, which was tagged by rs34536443/rs74956615, also colocalizes with the *cis* eQTL signal for *PDE4A* in artery tibial in GTEx v8 data (**Fig 4A**, $PP_{H4} > 0.99$ for colocalisation between the two traits and with the eQTL signal) where the severe COVID-19 risk allele, that is protective for SLE, is associated with reduced expression (**Table G in S1 Text**).

**Table 1. Association results for lead SNPs in the cross-trait meta-analysis/inverse meta-analysis of severe COVID-19 and SLE data.** Posterior possibilities (PP$_{H4}$) of colocalisation between signals were estimated with region ± 1 Mb of lead SNPs, if there are multiple independent signals the highest PP$_{H4}$ of colocalisation was shown. Predicted functional genes were inferred by lead SNP in the region and its LD with coding variants, eQTL data (GTEx v8 and eQTLGen), ENCODE ChIP-seq marker data, GeneHancer interactions data, and published functional studies.

**Signals showing aligned effect in a cross-trait meta-analysis**

| SNP | Position | A1 | meta | | SLE | | severe COVID-19 | | PP$_{H4}$ | predicted functional genes in SLE | predicted functional genes in COVID-19 |
|---|---|---|---|---|---|---|---|---|---|---|---|
| | | | OR (95% CI) | P | OR (95% CI) | P | OR (95% CI) | P | | | |
| rs7960611 | 12:10230416 | G | 1.14 (1.09–1.19) | 8.02 x 10$^{-09}$ | 1.11 (1.05–1.17) | 2.76 x 10$^{-04}$ | 1.17 (1.10–1.25) | 3.31 x 10$^{-07}$ | 95.4% | CLEC1A[1] | CLEC1A[1] |
| rs6869688 | 5:158883027 | G | 0.92 (0.90–0.95) | 4.57 x 10$^{-08}$ | 0.90 (0.87–0.94) | 1.79 x 10$^{-08}$ | 0.94 (0.91–0.98) | 1.96 x 10$^{-03}$ | 81.5% | IL12B[2] | IL12B[2] |
| rs10460393 | 2:198548306 | T | 1.09 (1.06–1.13) | 1.07 x 10$^{-09}$ | 1.11 (1.07–1.14) | 2.11 x 10$^{-08}$ | 1.08 (1.04–1.13) | 6.48 x 10$^{-05}$ | 80.7% | PLCL1[3], RFTN2[4] | PLCL1[3], RFTN2[4] |
| rs2431697 | 5:159879978 | C | 0.90 (0.87–0.92) | 8.51 x 10$^{-14}$ | 0.84 (0.81–0.87) | 2.12 x 10$^{-20}$ | 0.94 (0.91–0.98) | 2.42 x 10$^{-03}$ | 80.5% | MIR146A[5] | MIR146A[5] |
| rs4792891 | 17:43973498 | G | 0.90 (0.88–0.93) | 6.82 x 10$^{-10}$ | 0.92 (0.89–0.96) | 1.40 x 10$^{-05}$ | 0.90 (0.86–0.93) | 3.37 x 10$^{-08}$ | 56.3% | MAPT[6] | MAPT[6] |
| rs5022165 | 1:67788352 | A | 1.12 (1.08–1.16) | 9.10 x 10$^{-09}$ | 1.17 (1.12–1.23) | 1.82 x 10$^{-12}$ | 1.07 (1.02–1.13) | 8.52 x 10$^{-03}$ | 23.0% | IL12RB2[7] | IL12RB2[7] |
| rs3024897 | 2:191896564 | C | 0.87 (0.83–0.91) | 1.06 x 10$^{-08}$ | 0.82 (0.77–0.87) | 1.17 x 10$^{-10}$ | 0.91 (0.86–0.97) | 5.41 x 10$^{-03}$ | 14.6% | STAT1[8], STAT4[9] | STAT1[8], STAT4[9] |
| rs35605052 | 3:45916547 | T | 1.16 (1.12–1.21) | 2.14 x 10$^{-13}$ | 1.08 (1.03–1.13) | 6.95 x 10$^{-04}$ | 1.25 (1.18–1.32) | 1.02 x 10$^{-14}$ | 6.03% | CXCR6[10] | CXCR6[10], SLC6A20, FLT1P1, FYCO1, CCR1, CCR3 |
| rs7970893 | 12:113390679 | T | 0.92 (0.90–0.95) | 2.10 x 10$^{-08}$ | 0.94 (0.91–0.98) | 2.26 x 10$^{-03}$ | 0.90 (0.87–0.94) | 1.28 x 10$^{-07}$ | 0.00% | ALDH2, SH2B3 | OAS1, OAS2, OAS3 |

**Signals showing opposing effect in a cross-trait inverse meta-analysis**

| SNP | Position | A1 | meta | | SLE | | severe COVID-19 | | PP$_{H4}$ | predicted functional genes in SLE | predicted functional genes in COVID-19 |
|---|---|---|---|---|---|---|---|---|---|---|---|
| | | | OR (95% CI) | P | OR (95% CI) | P | OR (95% CI) | P | | | |
| rs11085727 | 19:10466123 | T | 1.21 (1.17–1.25) | 2.09 x 10$^{-31}$ | 0.80 (0.77–0.84) | 6.92 x 10$^{-27}$ | 1.19 (1.14–1.23) | 2.33 x 10$^{-18}$ | 99.3% | TYK2[11] | TYK2[11] |
| rs74956615 | 19:10427721 | A | 1.51 (1.39–1.63) | 2.19 x 10$^{-25}$ | 0.58 (0.53–0.65) | 2.75 x 10$^{-24}$ | 1.40 (1.29–1.53) | 3.04 x 10$^{-14}$ | 99.1% | PDE4A[12] | PDE4A[12] |
| rs1174683 | 1:183650428 | G | 1.16 (1.11–1.22) | 2.37 x 10$^{-09}$ | 0.82 (0.77–0.87) | 3.98 x 10$^{-10}$ | 1.12 (1.05–1.18) | 1.50 x 10$^{-04}$ | 60.1% | NCF2[13] | NCF2[13] |
| rs5778759 | 1:173328868 | C | 1.13 (1.09–1.16) | 3.32 x 10$^{-13}$ | 0.83 (0.80–0.86) | 1.98 x 10$^{-22}$ | 1.06 (1.01–1.11) | 9.66 x 10$^{-03}$ | 36.8% | TNFSF4[14] | TNFSF4[14] |
| rs17445836 | 16:86017663 | A | 1.14 (1.09–1.19) | 5.14 x 10$^{-10}$ | 0.83 (0.79–0.87) | 3.68 x 10$^{-16}$ | 1.07 (1.02–1.13) | 7.11 x 10$^{-03}$ | 35.7% | IRF8[15] | IRF8[15] |
| rs61811916 | 1:155045004 | C | 1.14 (1.09–1.19) | 4.11 x 10$^{-08}$ | 1.11 (1.05–1.17) | 4.59 x 10$^{-04}$ | 0.85 (0.80–0.91) | 4.86 x 10$^{-07}$ | 8.01% | ADAM15[16] | ADAM15[16], GBA, MUC1, THBS3, GBAP1 |

*(Continued)*

**Table 1.** (Continued)

| rs76073397 | 16:11386452 | C | 1.14 (1.09–1.19) | 3.21 x 10⁻⁰⁸ | 0.90 (0.84–0.95) | 5.14 x 10⁻⁰⁴ | 1.15 (1.09–1.22) | 9.67 x 10⁻⁰⁷ | 1.18% | *RMI2*[17], *CLEC16A, DEXI* | *RMI2*[17] |

Genes associated with autoimmune or infectious diseases includes: [1]experimental autoimmune encephalomyelitis (EAE). [2]psoriasis, crohn's disease (CD), inflammatory bowel disease (IBD), ankylosing spondylitis (AS), sclerosing cholangitis (SC), ulcerative colitis (UC), psoriatic arthritis (PsA), multiple sclerosis (MS), ulcerative colitis (UC), primary biliary cirrhosis (PBC), autoimmune thyroid disease (AITD), celiac disease (CeD), type 1 diabetes (T1D), juvenile idiopathic arthritis (JIA), rheumatoid arthritis (RA). [3]UC, SLE, CD, allergic rhinitis, asthma, RA, IBD, AS, psoriasis, SC. [4]atopic asthma. [5]SLE, Sjogren's syndrome (SS), RA, MS, AITD. [6]PBC, SS. [7]SLE, PBC, systemic scleroderma (SSc), RA, CD, MS, AS, IBD, behcet's disease (BD). [8]JIA, RA. [9]SLE, RA, SSc, PBC, SS, CeD, BD, IBD, autoimmune hepatitis type-1 (AIH), T1D, MS, AITD, CD, JIA, UC, non-typhoidal Salmonella bacteremia. [10]PBC, T1D, AIH, EAE. [11]SLE, COVID-19, psoriasis, SSc, MS, RA, T1D, PBC, AS, CD, SC, UC, IBD, AITD. [12]SLE, psoriasis, MS, JIA, T1D, AS, CeD, CD, UC, AITD. [13]SLE, RA, SSc, CeD. [14]SLE, asthma, atopic asthma, SSc, allergic rhinitis, AITD. [15]SLE, PBC, SSc. [16]SLE. [17]PBC, IBD, MS, CD, T1D, psoriasis, CeD, AS, UC, SC, JIA, asthma.

We found that the genetic associations in signal-B tagged by rs34725611 and rs11085727 colocalize with a *TYK2* eQTL signal in whole blood in eQTLGen [14] (**Fig 4B**) and GTEx v8 data, and adrenal gland in GTEx v8 data: PP$_{H4}$ > 0.98 for colocalisation between the two traits and with all eQTL signals (**Fig N in S1 Text**). eQTL summary statistics can be seen in **Table G in S1 Text**, where the associated allele effects can be compared across traits and eQTL. In all

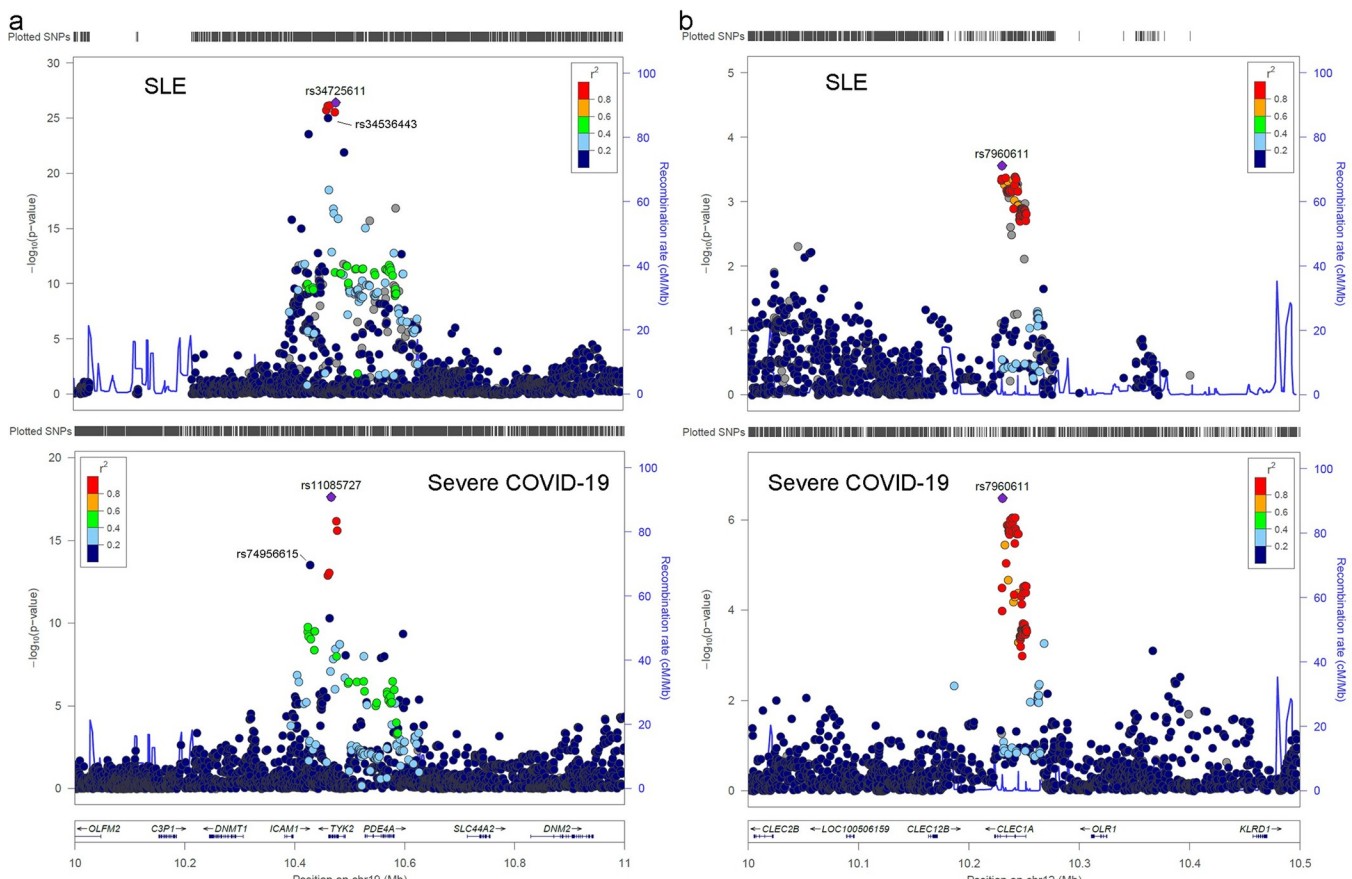

**Fig 3. Locus zoom plots across a)** *TYK2*, **b)** *CLEC1A* **for single marker associations with SLE, severe COVID-19.** The LD (r² in 1000 Genome project Phase 3 EUR) is identified by color.

**Table 2. *TYK2* association results for a) SLE and b) severe COVID-19 data, and c) summary of functional effects of associated alleles.** Independently associated SNPs in SLE and severe COVID-19 are displayed.

Table 2a. *TYK2* associations with SLE

| Signal number | SNP | position | A2 | A1 | freq(A1) | OR | 95% CI | P | bJ | bJ_se | pJ |
|---|---|---|---|---|---|---|---|---|---|---|---|
| Signal-A | rs34536443 | 19:10463118 | G | C | 0.034 | 0.53 | 0.47–0.59 | $9.80 \times 10^{-26}$ | -0.57 | 0.06 | $9.47 \times 10^{-20}$ |
| Signal-B | rs34725611 | 19:10477067 | A | G | 0.275 | 0.80 | 0.77–0.83 | $3.84 \times 10^{-27}$ | -0.16 | 0.02 | $2.86 \times 10^{-13}$ |

Table 2b. *TYK2* associations with severe COVID-19

| Signal number | SNP | position | A2 | A1 | freq(A1) | OR | 95% CI | P | bJ | bJ_se | pJ |
|---|---|---|---|---|---|---|---|---|---|---|---|
| Signal-A | rs74956615 | 19:10427721 | T | A | 0.047 | 1.40 | 1.27–1.55 | $3.04 \times 10^{-14}$ | 0.26 | 0.05 | $4.06 \times 10^{-08}$ |
| Signal-B | rs11085727 | 19:10466123 | C | T | 0.280 | 1.19 | 1.14–1.23 | $2.33 \times 10^{-18}$ | 0.14 | 0.02 | $2.83 \times 10^{-12}$ |

Table 2c. Functional effects of associated alleles in *TYK2 –PDE4A* locus.

| Signal number | Ref SNP | Minor/ancestral Allele | Ancestral Allele SLE effect | Ancestral Allele COVID Effect | Function | Gene | Ancestral Allele Functional Effect |
|---|---|---|---|---|---|---|---|
| A | rs34536443 | C/G; Ala1104Pro | Risk | Protective | Coding | Tyrosine Kinase 2 (*TYK2*) | Increased Gene function through increased phosphorylation [13] |
| A | rs34536443 | C/G | Risk | Protective | Regulation | Phosphodiesterase 4A (*PDE4A*) | Increased Gene Expression: eQTL data (**Table G in S1 Text**) |
| B | rs2304256 | A/C Phe362Val | Risk | Protective | Coding | Tyrosine Kinase 2 (*TYK2*) | Decreased Gene function though loss of exon 8 [15] |
| B | rs11085727 | T/C | Risk | Protective | Regulation | Tyrosine Kinase 2 (*TYK2*) | Decreased Gene Expression: eQTL data (**Table G in S1 Text**) |
| B | rs11085727 | T/C | Risk | Protective | Regulation | Serpin Family G Member 1 (*SERPING1*) | Increased Protein Expression: pQTL data [17] |
| B | rs11085727 | T/C | Risk | Protective | Regulation | C-X-C Motif Chemokine Ligand 10 (*CXCL10*) (IP-10) | Increased Protein Expression: pQTL data [17] |

* bJ, bJ_se, pJ: effect size, standard error and p-value from a joint analysis (multiple regression) of all the selected SNPs (results conditional on all other SNPs if selected from stepwise regression). † rs34536443 is in high LD with rs74956615 ($r^2 = 0.88$), rs34725611 is in high LD with rs11085727 ($r^2 = 0.97$). In table c) we refer to the common ancestral allele for effects where this is protective for severe COVID-19 and risk for SLE. Functional effects cover coding variation, *cis* acting gene transcript expression and *trans* acting protein product expression.

cases the protective allele for SLE, which is the risk allele for severe COVID-19, increases expression. However, signal-B is also associated with altered TYK2 function as a missense variant rs2304256 (V362F, exon 8), that is in strong LD ($r^2 = 0.98$ in SLE data) with rs11085727, acts as a splicing eQTL. The SLE protective allele promotes inclusion of exon 8 [15], which increases TYK2 function. Thus, signal-B provides conflicting results with respect to signal-A regarding the functional impact on TYK2. To understand the role of signals A and B on gene regulation, we studied the epigenetic landscape around these two association signals (**Fig O in S1 Text**). For signal A, there was evidence for localization to enhancer chromatin marks (H3K27Ac and H3K4Me1, **Fig P in S1 Text**). However, there was much less evidence for such alignment with signal-B (**Fig Q in S1 Text**). Signal-A is also observed to loop in 3D space to the promotor of *PDE4A* (**Fig R in S1 Text**). While we did observe other significant *cis* eQTLs with signal-B SNPs (**see Fig S in S1 Text**), none of them colocalized with COVID-19 or SLE signals ($PP_{H4} < 0.20$ in all cases).

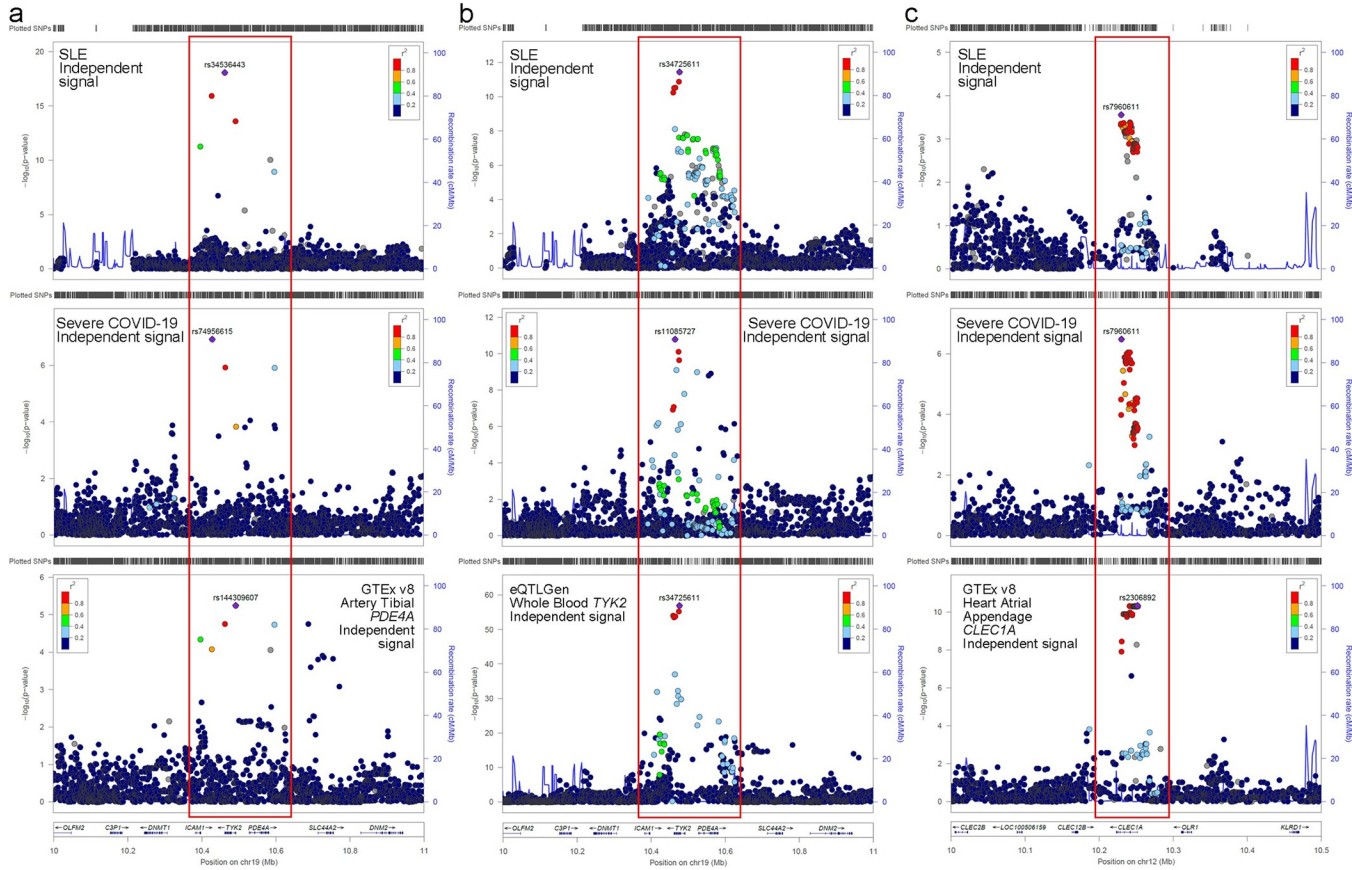

**Fig 4. Locus zoom plots across loci for marginal associations with SLE, severe COVID-19 and eQTL. a)** *PDE4A* **locus,** both diseases' signal-B colocalized with eQTL for *PDE4A*. **b)** *TYK2* **locus,** both diseases' signal-A colocalized with eQTL for *TYK2*. **c)** *CLEC1A* **locus,** both diseases colocalized with eQTL for *CLEC1A*. The LD ($r^2$ in 1000 Genome project Phase 3 EUR) is identified by color.

To explore the functional effects further, we looked for downstream effects of the shared *TYK2* associated SNPs on the expression levels of a set of 21 IFN-induced genes (dysregulated in SLE [16]) in human plasma proteome data [17]. Significant trans pQTLs (FDR < 0.01, **Table H in S1 Text**) for signal-B (rs11085727) were found for two targets: SERPING1 and CXCL10 (IP-10). Both proteins are induced by interferon and would be expected to require TYK2 activity for induction. The COVID-19 risk allele (T), that correlates with increased *TYK2* transcript expression, correlated with reduced amounts of SERPING1 and CXCL10 proteins in plasma (p = 0.0003).

## C-Type lectin domain family 1 member A (CLEC1A)

The meta-analysis identified a narrow peak of association between 10.2–10.3Mb on chromosome 12 that colocalized between the two traits (**See Fig 3B**; $PP_{H4}$ = 0.95 and **Table 3**, overlapped $N_{SNP}$ = 3,363). Both traits' association signals colocalized with eQTLs for *CLEC1A* in multiple tissues ($PP_{H4} \geq 0.97/0.87$ for eQTL colocalisation with COVID-19/SLE) in GTEx v8 data. **Fig 4C** displays the association in both diseases and eQTL data for heart (atrial appendage), see **Fig T in S1 Text** for the other eQTL colocalization. eQTL summary statistics can be seen in **Table G in S1 Text**. The risk allele for severe COVID-19 is also risk for SLE and is associated with reduced expression of *CLEC1A*. The lead variant rs7960611 is in LD with a missense variant rs2306894 ($r^2$ = 0.84).

**Table 3. *CLEC1A* association results for a) SLE and b) severe COVID-19 data, and c) summary of functional effects of associated alleles.** Independently associated SNPs in SLE and severe COVID-19 are displayed.

| Table 3a. *CLEC1A* associations with SLE | | | | | | | | | | | |
|---|---|---|---|---|---|---|---|---|---|---|---|
| Signal number | SNP | position | A2 | A1 | freq(A1) | OR | 95% CI | P | bJ | bJ_se | pJ |
| Signal-A | rs7960611 | 12:10230416 | A | G | 0.115 | 1.11 | 1.04–1.17 | $2.76 \times 10^{-04}$ | 0.10 | 0.03 | $2.77 \times 10^{-04}$ |

| Table 3b. *CLEC1A* associations with severe COVID-19 | | | | | | | | | | | |
|---|---|---|---|---|---|---|---|---|---|---|---|
| Signal number | SNP | position | A2 | A1 | freq(A1) | OR | 95% CI | P | bJ | bJ_se | pJ |
| Signal-A | rs7960611 | 12:10230416 | A | G | 0.124 | 1.17 | 1.11–1.24 | $3.31 \times 10^{-07}$ | 0.16 | 0.03 | $3.31 \times 10^{-07}$ |

| Table 3c. Functional effects of associated alleles in *TYK2 –PDE4A* locus. | | | | | | | |
|---|---|---|---|---|---|---|---|
| Signal number | Ref SNP | Minor/ancestral Allele | Ancestral Allele SLE effect | Ancestral Allele COVID Effect | Function | Gene | Ancestral Allele Functional Effect |
| Signal-A | rs2306894 | C/G; Gly26Ala | Protective | Protective | Coding | C-type lectin domain family 1 member A (*CLEC1A*) | Gly26Ala No publication found |
| Signal-A | rs7960611 | G/A | Protective | Protective | Regulation | C-type lectin domain family 1 member A (*CLEC1A*) | Increased Gene Expression: eQTL data (**Table G in S1 Text**) |

* bJ, bJ_se, pJ: effect size, standard error and p-value from a joint analysis (multiple regression) of all the selected SNPs (results conditional on all other SNPs if selected from stepwise regression). In table c) we refer to the common ancestral allele for effects where this is protective for severe COVID-19 and SLE. Functional effects cover coding variation and *cis* acting gene transcript. † rs2306894 is in high LD with rs7960611 ($r^2 = 0.84$).

## Discussion

Our results indicate that there are shared genetic effects between the autoimmune disease SLE and the clinical consequences of COVID-19. The locus with the most evidence of shared effects was the Janus kinase (JAK), *TYK2*, that promotes IL-12 and IFN-I signaling. Here there are two separate genetic association signals (designated A and B) shared between severe COVID-19 and SLE. Importantly for both, the genetic factors for SLE risk mitigate the outcome following SARS-Cov2 infection. In seeking to uncover the mechanisms underlying these relationships it was apparent that the functional effects of the risk alleles are complex. Signal-A at *TYK2* is likely driven by a coding P1104A variant (rs34536443) whose COVID-19 risk allele has been shown to impair TYK2 target phosphorylation [13]. This is further supported by the therapeutic effect of a TYK2 inhibitor in psoriasis [18], and by observed risk in other infectious disease such as tuberculosis where it has been found that homozygosity for the minor allele (C) of rs34536443 is risk, in line with severe COVID-19, and strongly impairs IL-23 signaling in T cells and IFN-γ production in PBMC [19,20]. Signal-A, led by rs34536443, was also found to colocalize with an eQTL for nearby *PDE4A*, which encodes a phosphodiesterase that regulates cAMP. This enzyme has multiple potential roles, however PDE4A inhibitors have been shown to have anti-inflammatory activity and are being studied in AID and inflammatory lung diseases [21]. The severe COVID-19 risk alleles are associated with decreased expression of *PDE4A*, while they are protective for SLE. The *PDE4A* eQTL cell type is heterogeneous however and the relevance to SLE is unclear. Signal-B includes another missense variant in *TYK2*, namely rs2304256 (V362F) in exon 8, but this also acts as a splicing mutation and the missense variant is missing from the spliced transcript. The severe COVID-19 risk allele promotes inclusion of exon 8 in *TYK2* that is essential for *TYK2* binding to cognate receptors [15]. Therefore

signal-B comprises evidence for two functional effects with respect to COVID-19 risk alleles, one of which increases function of *TYK2* through altered splicing (rs2304256 (V362F)) and one that is correlated with increased expression of *TYK2* (rs11085727). It may be that the overall reduction of *TYK2* activity caused by the COVID-19 risk alleles in signal-A evokes a compensatory effect on overall gene expression, which is designed to mitigate the deleterious effect of the missense variants–an example of regulatory variants modifying the penetrance of coding variants [15,22]. This conjecture is supported by the lack of epigenetic marks in the signal-B region of *TYK2*.

The severe COVID-19 risk allele for signal-B at *TYK2* is associated with reduced SERPING1 and CXCL10 protein expression, implying that the minor allele at signal-B in the *TYK2* locus reduces some aspect of *TYK2* function. CXCL10 (IP-10) is a chemokine that acts on Th1 cells and is key regulator of the cytokine storm immune response to COVID-19 infection [23]. SERPING1, an inhibitor of complement 1 (C1-inh), is known to be reduced by infection and this reduction correlates with more severe COVID-19 [24]. Therefore genetic predisposition to low SERPING1 expression may increase risk for COVID-19 through the same dynamics as reduced levels due to infection. This and the effect of reduced levels of CXCL10 are likely just two examples of altered IFN induced activity that affects risk for disease.

We found agreement in direction of effect of association in *CLEC1A*. *CLEC1A* is interesting as C-type Lectin receptors are involved in fungal recognition and fungal immunity. Genetic variation in *CLEC1A* is a risk factor for the development of Aspergillosis in immunosuppression [25]. CLEC1A is a negative regulator of dendritic cells [26]. Therefore the SLE and severe COVID-19 risk allele, being associated with reduced expression of *CLEC1A*, would be expected to exert a pro-inflammatory effect. We also found agreement in direction of effect of associations in 3 other loci (*IL12B*, *PLCL1-RFTN2*, *MIR146A*) that showed relatively strong evidence of colocalization. The modest p-values and relatively high colocalisation possibilities support them as good candidates to follow up in larger studies. At both *IRF8* and *TNFSF4* the evidence for association in severe COVID is moderate yet the signals do show some evidence of colocalizing with opposing effects in SLE. With prominent roles in the pro-inflammatory IFN response these two loci should be a focus when larger data in severe COVID-19 are available. *IRF8* provides more evidence that the IFN pathway is important in the balance between SLE risk and infection as mutations that impair *IRF8* transcriptional activity have been found to cause immunodeficiency [27]. Interferons constitute one of the main means of host defense against viruses and hence have been well studied in the context of COVID-19 [28–30]. In SLE, evidence for interferon activity is present in about half of the patients and is often present in those with more severe disease [31–33]. Although elevated interferon has been implicated in other AID, the role is prominent in SLE. This has been exploited with therapeutic agents designed to antagonize type I interferon activity showing benefit in SLE [34]. Parallels between SLE and viral infection extend beyond interferon activation though. As stated above there are SLE risk genes that act in the intracellular viral sensing pathways. SLE is characterized by an immune response against host nucleic acids. The means by which the immune system loses tolerance to these structures appears to involve aberrant exposure of self through the pathways that are designed to sense foreign nucleic acids, as happens during viral infection [35]. Further investigation into the genetic correlation between SLE and severe COVID-19 will help explain the genetic basis of both diseases, which may be in part due to variation in response to viral infection. Risk alleles for SLE, that are also risk for severe COVID-19, may persist in the population due to protective effects against other exposures such as fungal infection. The opposing effects we find at the *TYK2* locus is compatible with the hypothesis that there are alleles in the general population that, while represent a risk for SLE, persist possibly due to an innate immune protection against pathogens [36–41] including viruses.

## Material and methods

### Data for genome-wide and local genetic correlation

Full summary-level GenOMICC release 1 data were downloaded from https://genomicc.org/data. These data resulted from a GWAS of 1,676 critically ill patients with COVID-19 (severe COVID-19) of European ancestry from 208 UK intensive care units (GenOMICC GWAS data release 1), and ancestry-matched control individuals (8,380 of European ancestry) selected from the large population-based cohort of UK Biobank [1]. Controls with a known positive COVID-19 test were excluded [1]. An SLE meta-analysis of three previously published European GWASs was used (the SLE main cohort [42], 4,036 cases and 6,959 controls; the Genentech cohort [2], 1,165 cases and 2,107 controls; the SLEGEN cohort [43], 533 cases and 2,543 controls), each of these data have been pre-phased (SHAPEIT [44]) and imputed (IMPUTE [45,46], 1000 Genomes phase 3 [47]) using the same pipeline as in the previous studies of these data where they were imputed to the 1000 genomes phase 1 density [4,48]. SNPTEST was run in each dataset using principal components as covariates to control for population structure as in the original studies. A standard fixed effects inverse variance approach was used for meta-analysis using our own scripts written in R, that also checked for allele matching and strand issues, and METAL [49]. The genomic inflation factor ($\lambda_{1000}$) [50] was 1.02. To evaluate genetic correlation between SLE and severe COVID-19, we used conventional cross-trait LD score regression (LDSC) [51,52] to calculate genome-wide genetic correlation ($r_g$). All the overlapping SNPs between the SLE meta-analysis and the COVID-19 GenOMICC European data were retained for use. The number of SNPs were reduced to common SNPs (MAF > 0.01) from the European 1000 Genomes populations [47] ($N_{SNP}$ = 413,464 genome-wide) as these data were used as the LD reference panel in the genetic correlation analyses.

To increase power for local genetic correlation detection, while maintaining the same ancestry as required by the methodology [53], we added SLE Immunochip data [5] from a previous study (3,568 cases and 11,245 controls independent of the three European GWAS) to the SLE meta-analysis. These data were also imputed to the density of the 1000 Genomes Phase 3 data. The new meta-analysis also used a standard fixed effects inverse variance approach (MAF > 0.01 and INFO > 0.9). The genomic inflation factor ($\lambda_{1000}$) was 1.03. Local genetic correlation was performed using a recent approach that uses summary statistics ($\rho$-HESS) [53] to estimate local SNP-level heritability and genetic covariance (correlation).

### Data for the SLE–severe-COVID-19 meta-analyses, cross disease colocalisation analyses and fine-mapping

To maximize power for genetic association [54] we obtained multi-ancestry severe COVID-19 vs. population genetic association data from round 6 of the COVID-19 Host Genetics Initiative (COVID-19 hg, https://www.covid19hg.org/) where the GenOMICC study was a subset of these data [55]. The severe COVID-19 phenotype is defined as individuals critically ill with COVID-19 based on either requiring respiratory support in hospital or who died as a consequence of the disease [55]. These data comprised 8,779 cases vs. 1,001,875 controls (A2_ALL_-leave_23andme) [55] and were obtained from a google storage bucket provided by COVID-19 hg. The association summary data was the result of a meta-analysis of 60 studies from 25 countries and was performed by the provider with fixed effects inverse variance weighting after filtering for allele frequency > 0.001 and imputation INFO > 0.6 applied to each study. The SLE meta-analysis that included Immunochip data [5] was used for the meta-analysis with severe-COVID-19 and for fine-mapping (genome-wide overlapped $N_{SNPs}$ = 1,559,546). We also used

the African American (2,970 cases and 2,452 controls) and Hispanic (1,872 cases and 2,016 controls) samples from the Immunochip study [5] for replication (see Supplementary information).

### SLE–severe-COVID-19 meta-analyses

We performed two cross-trait meta-analyses between the SLE meta-analysis (Three EUR GWAS + EUR Immunochip) summary statistics and the severe COVID-19 HGI release 6 GWAS summary statistics using R, that also checked for allele matching and strand issues, and METAL [49]. Firstly, both diseases' summary statistics were analyzed using the inverse variance approach (upper plot in **Fig 2**). A second analysis was undertaken in which the severe COVID-19 direction of effect was reversed followed by a standard inverse variance meta-analysis (lower plot in **Fig 2**). This second approach is more powerful to detect areas of the genome that have genetic association with both diseases but the direct of effect is opposing between SLE and severe COVID-19. In both meta-analyses we only retained and plotted p-values for SNPs that had p < 0.01 in both diseases and had shared direction of effects with respect to each of the two types of meta-analysis. Any SNPs that passed a significance threshold of $P < 5 \times 10^{-08}$ in meta-analysis in both traits were considered as candidates for shared association. These were followed up by fine mapping in both traits and colocalisation analysis. The MHC region was not included.

### Checking published and candidate associated loci across traits for shared association loci

Loci published as associated in each trait were checked for locus-wide association with the other trait ($p < 1 \times 10^{-05}$) in our summary association data. Loci were defined as the lead published SNP +/-1mb. Candidate shared loci were visible inspected using locus-wide LocusZoom plots [56] and loci were checked for colocalisation of association between the two diseases. We also investigated any locus that had a SNP with $p < 1 \times 10^{-05}$ in both traits in our summary data for colocalization, where the loci were defined as the shared associated SNP +/-1mb. On both these analyses we only declared a locus as shared if the colocalisation probability was greater than 0.9.

### eQTL data

The cis-eQTL summary statistics data was obtained from eQTLGen Consortium (https://www.eqtlgen.org/) [14], which includes eQTL data from 31,684 whole blood samples across 37 cohorts cohorts mainly of European origin, and from European specific eQTL data from GTEx v8 across 54 tissues [57].

### pQTL data

Two studies' combined summary pQTL data [58,59] were downloaded from https://gwas.mrcieu.ac.uk. These two studies data were combined and analyzed previously [17]. We focused our pQTL analysis on 21 IFN induced genes previously defined [16]. Associations between the SNPs in our study and the 21 gene's expression were retrieved if included in the study, otherwise tagging SNPs were used. A Bonferroni adjustment was made for multiple testing across all SNP/gene combinations.

### Fine-mapping

Our main fine-mapping analysis consisted of comparing summary association data between SLE and severe COVID-19. This consisted of an approximate stepwise regression using COJO [60] in both diseases' data to identify independent signals and colocalization analyses to

investigate whether shared association were coincidental. For supplementary information we also ran stepwise regression on the SLE individual level data, which we were unable to do on the COVID-19 data and so no comparison could be made.

**COJO.**  To find independently associated variants using summary data, we performed an approximation of stepwise regression using GTCA 1.93 [61] (COJO [60], 'cojo-slct'). Lead SNPs from stepwise regression were taken as index SNPs and marginal signals were obtained by conditional analysis (-cojo-cond) on the set of index SNPs that were not in the signal of interest. The parameters for stepwise selection were p-value $< 1 \times 10^{-5}$, a collinearity cutoff of 0.9 and a distance of 10Mb. The SLE main cohort controls were used as the reference panel of SNPs to estimate LD.

**Colocalisation.**  For each of the loci we found to have shared association between SLE and severe COVID-19, we used coloc [7] to perform locus-wide genetic colocalisation analysis. This returns the posterior probability that the two diseases share the same causal variant(s) in the region. We used standard coloc that assumes one casual variant and applied this to the marginal signals obtained using COJO. The SLE main cohort controls were used as the reference panel of SNPs to estimate the LD. GTEx v8 summary statistics across all tissues and eQTLGen whole blood summary statistics were used for the analysis.

For both the SLE/COVID-19 and the disease/eQTL colocalisation, signals were deemed to colocalize if: (1) when setting the prior probability of a SNP as associated with both traits $= 5 \times 10^{-05}$ ($p_{12}$, default $= 1 \times 10^{-05}$), the posterior probability of colocalisation ($PP_{H4}$) $> 0.5$ and (2) when setting $p_{12} = 1 \times 10^{-5}$, the posterior probability of different causal variants ($PP_{H3}$) $< 0.5$ [62].

## Haplotype analysis

Conditional haplotype-based association testing was performed on cases and controls in the European main SLE GWAS data and those with European/Hispanic/African American ancestry from the SLE Immunochip study using Plink [63]. An Independent effect for rs11085727 and rs2304256 was tested on the background of all the potential lead SNPs including rs34536443, rs74956615, rs11085727, rs2304256, rs12720356, rs280497, rs12720358 by using the PLINK '—hap-snps' command on the full set of SNPs with the '—independent-effect' option on [rs11085727, rs2304256] with '—chap'. Block estimations were performed within 200 kb. All variants with MAF $< 0.001$ were removed. The range of the 90% D-prime confidence interval was 0.70–0.98. The upper level for the confidence interval for historical recombination was 0.90 and strong LD pairs fraction was equal to 0.95.

## Stepwise regression on individual level SLE data

In supplementary data analysis only, we analyzed the SLE GWAS individual level data for association using SNPTEST [45] fitting an additive model. This analysis was performed with three European SLE GWAS and the Immunochip data using SNPs with an imputation info score of $> 0.7$ and MAF $> 0.01$. Each dataset was included as a separate cohort in SNPTEST with covariates including principal components (PC1-3) for population structure and a discrete covariate for study. The same effect was assumed for all studies, as with a fixed effect meta-analysis. The results from the single-marker analysis using this approach were similar to those from the standard meta-analysis on the summary data (compare results for rs34536443 in **Tables F** and **I in S1 Text** for example) and the top associated SNPs was the same. We then ran forward stepwise selection by adding the top SNPs at each stage as a covariate to identify independently associated variants. This can be referred to as a one-stage approach to a meta-analysis using individual level data [64]. An alternative approach would be to use the top SNP at each step as a covariate in a regression analysis of each study separately and then meta-

analyze the results at each step (a two-stage approach). This would then rely on the meta-analysis approximation and not allow for an easy derivation of marginal signals when multiple independent associations are obtained.

### Epigenetic modification and chromatin looping

Epigenetic modification and chromatin looping information were taken from resources available at the UCSC genome browser (http://genome.ucsc.edu). Enrichment of modifications to histone proteins (layered H3K27Ac, H3K4Me1, and H3K4Me3 track sets) determined by a ChIP-seq assay were from the ENCODE Consortium. Common dbSNP153 data (1000 Genomes phase 3, MAF > 0.01) was used, associated variants were highlighted. A highly filtered "double elite" subset of regulatory elements (including enhancers and promoters) and their inferred target genes in the plotting region were added as track sets, data was provided by the GeneHancer database [65].

### Network and pathway enrichment analysis

Molecular interactions were obtained from the STRING 11.5 database [66]. All the potentially shared SLE and severe COVID-19 associated genes from the cross-trait meta-analysis (Table 1) were mapped to the whole network, connected nodes are shown in **Fig M in S1 Text** with indication of the type of interaction evidence. Gene Ontology (GO) process [67], local STRING network clusters [66], WikiPathways [68], KEGG pathway classification [69], Reactome Knowledgebase [70], and DISEASES database [71] were used for pathway enrichment of all the potentially shared genes.

## Supporting information

**S1 Text. Supplementary Results.**
(DOCX)

## Acknowledgments

We thank Nick Dand for reviewing draft versions of this paper.

## Author Contributions

**Conceptualization:** Yuxuan Wang, Deborah S. Cunninghame Graham, David L. Morris, Timothy J. Vyse.

**Data curation:** Yuxuan Wang, Suri Guga, Kejia Wu, Phil Tombleson, Mary E. Comeau, Carl D. Langefeld, David L. Morris, Timothy J. Vyse.

**Formal analysis:** Yuxuan Wang, Suri Guga, Kejia Wu, Zoe Khaw, Konstantinos Tzoumkas, Phil Tombleson, Mary E. Comeau, Carl D. Langefeld, David L. Morris, Timothy J. Vyse.

**Funding acquisition:** David L. Morris, Timothy J. Vyse.

**Investigation:** Yuxuan Wang, Deborah S. Cunninghame Graham, David L. Morris, Timothy J. Vyse.

**Methodology:** Yuxuan Wang, David L. Morris, Timothy J. Vyse.

**Project administration:** Deborah S. Cunninghame Graham, David L. Morris, Timothy J. Vyse.

**Resources:** David L. Morris, Timothy J. Vyse.

**Supervision:** Deborah S. Cunninghame Graham, David L. Morris, Timothy J. Vyse.

**Validation:** Yuxuan Wang, David L. Morris, Timothy J. Vyse.

**Visualization:** Yuxuan Wang, David L. Morris, Timothy J. Vyse.

**Writing – original draft:** Yuxuan Wang, Deborah S. Cunninghame Graham, David L. Morris, Timothy J. Vyse.

**Writing – review & editing:** Yuxuan Wang, Deborah S. Cunninghame Graham, David L. Morris, Timothy J. Vyse.

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
