## [Decision Letter · Decision Letter 0]

13 Jul 2022

Dear Dr Morris,

Thank you very much for submitting your Research Article entitled 'COVID-19 and Systemic Lupus Erythematosus genetics: a balance between autoimmune disease risk and protection against infection' to PLOS Genetics.

The manuscript was fully evaluated at the editorial level and by independent peer reviewers. The reviewers appreciated the attention to an important problem, but raised some substantial concerns about the current manuscript. Based on the reviews, we will not be able to accept this version of the manuscript, but we would be willing to review a much-revised version. We cannot, of course, promise publication at that time.

If you decide to revise the manuscript for further consideration at PLOS Genetics, please aim to resubmit within the next 60 days, unless it will take extra time to address the concerns of the reviewers, in which case we would appreciate an expected resubmission date by email to plosgenetics@plos.org.

[LINK]

We are sorry that we cannot be more positive about your manuscript at this stage. Please do not hesitate to contact us if you have any concerns or questions.

Yours sincerely,

Giorgio Sirugo

Associate Editor

PLOS Genetics

Scott Williams

Section Editor: Human Variation

PLOS Genetics

Reviewer's Responses to Questions

**Comments to the Authors:**

Reviewer #1: In this study, the authors assessed genetic overlap between systemic lupus erythematosus and severe COVID-19 by performing a genetic correlation analysis and a trans-trait meta-analysis. They found that both disorders are genetically correlated and identified several loci showing evidence of shared association between them. Overall, this is a nice study that brings relevant information. The manuscript is pleasant to read. Nevertheless, some points should be considered by the authors.

- In the methods section, authors described two different COVID-19 datasets, full summary-level data of 2,244 critically ill patients with COVID-19 from 208 UK intensive care units and a very severe respiratory confirmed COVID vs. population round 6 data of 8,779 cases vs. 1,001,875 controls. However, they only referenced the second one in the results section. How many COVID-19 dataset were included in the analysis, one or two? For which step of the study was each of them used? Where do the controls of the UK cohort come from? Have the results of these two GWAS been published? If so, the corresponding references should be added. Please, clarify these points.

- In the methods section was reported that “Cases and controls from the European (EUR) genetic ancestry group were used in genetic correlation studies” but it is not described how many individuals were from European origin.

- A more detailed description of how meta-analysis of the different SLE datasets was carried out should be included.

- Please, clarify how many SNPs overlapped between the SLE meta-analysis and the COVID-19 data.

- Results of the Severe COVID-19 - SLE meta-analyses, which are included in the supplementary material, should be moved to the main text (including supplementary table 1).

- Authors stated: “further examination of regions driving this correlation identified multiple loci with both positive and negative correlation”. What other loci, in addition to TYK2 and CLEC1A, correlated between both diseases? Did any of these loci overlap with the shared loci identified in the meta-analysis? In order to clarify this issue, a table including the observed local genetic correlations and their p-values could be added.

- In the results section, authors stated: “this highlighted suggestive evidence of shared association at 13 other loci”. However, in Supplementary Table 1 all the reported associations reached the established significant threshold, why are they considered suggestive associations then?

- It is not clear why the authors focus the study on TYK2 and CLEC1A. Besides these two genes, there are other shared associations that would be interesting to discuss since they map in immune-related loci, such as IL12B (which indeed showed strong evidence for colocalisation of shared signals), IL12RB2, MIR146A, STAT1/STAT4…

- In addition, it should be very interesting to highlight in supplementary table 1 which of the shared signals have been previously associated with SLE and/or COVID-19 as well as with other autoimmune or infectious diseases.

Reviewer #2: “In the manuscript "COVID-19 and Systemic Lupus Erythematosus genetics: a balance between autoimmune disease risk and protection against infection", Wang and colleagues combine genotypic data from four large-scale association studies of SLE in individuals of European ancestry (including three GWAS), then do a meta-analysis of this SLE data with large-scale association data from “very severe respiratory” COVID-19 from individuals of European ancestry. The authors then compare the genetic associations between both traits (i.e., SLE and “severe COVID”) by looking at correlations and shared associations between loci. To help prioritize and understand the regulatory effects of the associated variants, in silico analyses integrating publicly available cis-eQTL and pQTL data, fine mapping and colocalisation analysis, and integration with regulatory data (such as epigenetic marks available at the UCSC web browser) were computed. This study reports a few shared loci with alleles with both positive and negative correlations between the traits.

The study is original and interesting, but poorly written. Specifically, the evidence of shared loci with alleles with both positive and negative correlations between the traits is significant and novel. However, the brief and vague description of the merging of genotypic data from different studies hinders an assessment of the quality of the results. Details required to allow several analyses to be reproduced are not described. Neglect in the writing is evident through the limited acknowledgment of previous literature, the lack of an explanation of the purpose and the analyses conducted prior to reporting results, the abundance of acronyms that are not defined, and the lack of clarity of several statements.

Below I offer several recommendations to help improve the clarity of this manuscript.

Major recommendations:

1. Prior to reporting results, please explain what the research question (i.e., the goal) was, and summarize the approach, or methods used to achieve that goal. This applies to the Results section. It also applies to the Abstract; as written, the meaning of the second sentence (“We find that severe COVID-19 and Systemic Lupus Erythematosus (SLE) are genetically correlated”) isn’t clear, the reader does not know what the authors mean by “genetically correlated”. Thus, explaining the goals and approach prior to the results will help the readers understand and clarify the text.

2. In the Methods section, the brief description of the analysis “combining the genotype data on the three European SLE GWASs and the Immunochip data” is worrisome. Ideally, the data from these different studies would have been merged by meta-analysis; for merging of the genotypic data, details of quality control measures need to be described. The brief, single statement that “Covariates included principal components for population structure and a discrete covariate for study” does not allow the reader to assess quality control and the robustness of the results. For example, what is the inflation factor (lambda) of the merged dataset?

3. Both the “Haplotype analysis” and the “Epigenetic modification and chromatin looping” sections of the Methods are too brief and vague, and should describe the details of the analyses computed to allow the analyses to be reproduced.

4. A description of how “severe COVID” was clinically defined is not provided. The clinical criteria for “very severe respiratory confirmed COVID” need to be described.

Other recommendations:

1. In the Abstract, the statement that “Our analyses suggest that (…) some SLE risk alleles may persist in the population due to protection from viral infection” is not backed up by experiments conducted in this study. Instead, other studies have shown that SLE risk alleles are protective against infection. Please clarify.

2. Please use capital letters appropriately. For ex, “Systemic Lupus Erythematous” and “Autoimmune Disease” shouldn’t be capitalized. In the Supplementary Data, “east Asians populations” should be “East Asian populations” (lines 57 and 63).

3. Instead of “Type-1 interferon pathway” or “IFN-1 signaling”, please replace the Arabic by the roman character, that is “type I interferon pathway” or “IFN-I”.

4. Please define all acronyms. These include “meta” in the Methods, “e-gene QTL” in the Discussion, and “RA”, “T1D”, and “1kg” in the Supplementary Data section.

5. In the Results, the inclusion of Bentham et al (2015) as the sole citation for the statement “The TYK2 locus has previously been found to be associated with SLE” is biased. Multiple studies have implicated TYK2 since 2005 (starting with Sigurdsson et al, Am J Hum Genet 2005).

6. In the Results, the sentence “The lead SLE SNP rs34536443 for signal-A is a missense variant and the SLE protective allele (C) has been found to reduce TYK2 function (ref #8)” isn’t clear. Please clarify, as reported in ref #8, that “minor allele homozygosity at rs34536443 drives a near complete loss of TYK2 function and consequently impairs type I IFN, IL-12 and IL-23 signaling”.

7. In Table 1, please convert the beta and SE to OR and CI.

8. In the Discussion, the statement that “In SLE, evidence for interferon activity is present in about two thirds of patients” is not accurate, as those references report that “about half of the patients studied showed dysregulated expression of genes in the IFN pathway” (Baechler et al, 2003; also Kirou et al, 2004), and that “41% of patients expressed high levels of IFN-inducible genes (Kirou et al, 2005).

9. In the Discussion, I would not use the word “unusual” in this sentence: “SLE is unusual among AID in that it is characterized by an immune response against host nucleic acids”, as antinuclear antibodies are found in other AID such as rheumatoid arthritis, systemic sclerosis, or Sjogren’s syndrome.

10. There are multiple sentences in the Discussion whose message is not clear. I recommend trying to clarify the meaning of the following sentences: “the signal comprises both this splicing eQTL as well as an e-gene QTL for TYK2 that, with respect to the COVID-19 risk allele, elevates gene expression”; “SERPING1 is an inhibitor of complement 1 (C1-inh) known to be reduced by infection which correlates with more severe COVID-19”; “It could be that genetic predisposition to low SERPING1 expression increases risk for COVID-19 through the same dynamics as reduced levels due to infection”; “CLEC1A is a negative regulator of dendritic cells, because the SLE and severe COVID-19 risk allele is associated with reduced expression of CLEC1A, it would be expected to exert a pro-inflammatory effect”.

11. In the Discussion, I feel that the sentence “The genetic correlation between SLE and severe COVID-19 will therefore help illuminate the genetics behind variation in response to viral infection” is an overstatement, and should either be clarified or toned down.

12. I’m not convinced that ref #30 explains the “balance between robust immune response and risk for AID”. In this exome-wide association study of psoriasis, Dand et al (2017) do mention “the hypothesis that the common ancestral alleles of IFIH1 and TYK2 contribute to a robust immune response to pathogens, but this comes at the expense of increased risk of immune-mediated disease.” However, as they mention, this is a known, previously formulated hypothesis, and not an hypothesis formulated by this study.

Reviewer #3: Wang et al. focused on the genetic features of COVID-19 and SLE. However, although the idea is novel and interesting, the author did not provide sufficient justification for why these two traits, especially since recent studies have shown no higher rate of severe COVID-19 in patients with SLE; rather the most severe outcome is due to comorbidities or untreated SLE. (PMID: 35172961) Although a recent survey showed patients with SLE do have a lower serological response to the vaccine, this was identified to be associated with several types of medicinal uses. Another recent study found that most patients with SLE and confirmed COVID-19 were able to produce and maintain a serological response despite the use of a variety of immunosuppressants (PMID: 34075358)

Authors used Coloc to examine shared causal SNPs of traits (COVID-19 and SLE). However, this does not establish a causal relationship between the traits. The author could consider using Mendelian randomization to identify biological mediators in the causal pathways using GWAS and pQTL results that are available in the public domain.

Line 30: the COVID GWAS study was conducted as a trans-ethnic study. The controls were matched to the cases for genetic ancestry and other factors. The three SLE GWAS used only European ancestry individuals. The authors should address the population discrepancy in performing a meta-analysis and wehether any of the variants identified have higher frequency in certain populations.

Ref 2: North Americans of European descent

Ref 3: women of European ancestry

Ref 4: mainly southern European ancestry

Line 58: cis-eQTL signals for PDE4A in what cell types? And is this cell type relevant to SLE?

Line 78: the authors looked into the functional effects of SNPs at the TYK2 locus. The authors selected a set of 21 IFN-induced genes that are upregulated in SLE in the pQTL dataset published by Zhen et al. Although two proteins, SERPING1 and CXCL10, were both found to have a significant trans-pQTL, the authors did not offer a table with the summary statistics. It’s also unclear why the authors limited the pQTL search only to the subset of the 21 IFN-induced genes, as there may be other interesting potential pathways involved in the response of the risk alleles. This could be an excellent opportunity to decipher the conflicting results seen in signal-A and signal-B.

Line 84: authors performed a trans-trait meta-analysis. The idea is interesting; however, it did not address the validity of the results. It is well known that mixing dissimilar studies results in reduced effectiveness. COVID and SLE may have overlapping genomic features, but there is no evident similarity in disease etiology. That being said, I do not have high confidence in the findings of CLEAC1A and the analysis presented.

**Have all data underlying the figures and results presented in the manuscript been provided?**

Reviewer #1: Yes

Reviewer #2: **No: **Summary statistics are provided under Supplementary Data in PDF format, not in spreadsheet format.

Reviewer #3: Yes

PLOS authors have the option to publish the peer review history of their article (what does this mean?). If published, this will include your full peer review and any attached files.

Reviewer #1: **Yes: **Ana Márquez

Reviewer #2: **Yes: **Paula Sofia Ramos

Reviewer #3: No

---

## [Decision Letter · Decision Letter 1]

18 Sep 2022

Dear Dr Morris,

We are pleased to inform you that your manuscript entitled "COVID-19 and systemic lupus erythematosus genetics: a balance between autoimmune disease risk and protection against infection" has been editorially accepted for publication in PLOS Genetics. Congratulations!

Yours sincerely,

Giorgio Sirugo

Academic Editor

PLOS Genetics

Scott Williams

Section Editor

PLOS Genetics

Comments from the reviewers (if applicable):

Reviewer's Responses to Questions

**Comments to the Authors:**

Reviewer #1: I have no additional comments.

Reviewer #2: The revised manuscript includes the suggested analytical details and clarification of the text.

As a minor note, it seems that something is missing on line 551, where it states “(details here)”.

I have no further suggestions.

**Have all data underlying the figures and results presented in the manuscript been provided?**

Reviewer #1: Yes

Reviewer #2: Yes

PLOS authors have the option to publish the peer review history of their article (what does this mean?). If published, this will include your full peer review and any attached files.

Reviewer #1: **Yes: **Ana Márquez

Reviewer #2: No

**Data Deposition**

http://datadryad.org/submit?journalID=pgenetics&manu=PGENETICS-D-22-00576R1

**Press Queries**

---

## [Editor Report · Acceptance letter]

11 Oct 2022

PGENETICS-D-22-00576R1 

COVID-19 and systemic lupus erythematosus genetics: a balance between autoimmune disease risk and protection against infection 

Dear Dr Morris, 

We are pleased to inform you that your manuscript entitled "COVID-19 and systemic lupus erythematosus genetics: a balance between autoimmune disease risk and protection against infection" has been formally accepted for publication in PLOS Genetics! Your manuscript is now with our production department and you will be notified of the publication date in due course.

With kind regards,

Anita Estes

PLOS Genetics

On behalf of:
